# Rural Population Aging, Capital Deepening, and Agricultural Labor Productivity

**Danhong Shen, Haimeng Liang *** and Wangfang Shi

School of Economics and Management, North University of China, Taiyuan 030051, China;
danhongshen@nuc.edu.cn (D.S.); d1378021048@163.com (W.S.)
* Correspondence: lianghaimeng2021@163.com; Tel.: +86-187-3475-6498

**Abstract:** Whether the aging of rural population will affect the high-quality development of China's agriculture is an important issue facing the construction of China's characteristic modern agriculture. Using panel data from 30 provinces in China from 1999 to 2020, this paper used fixed effects models and mediation models for econometric regressions to explore the relationship between rural population aging and agricultural development, as well as the intermediate mechanisms involved. The study found that there was a positive relationship between rural population aging and agricultural labor productivity, and aging can stimulate agricultural development. Mechanism analysis revealed that rural population aging can promote the adjustment of the structure of agricultural factors, drive the deepening accumulation of capital and other factors, promote the modernization of agricultural production methods, and ultimately achieve an increase in agricultural labor productivity. Further examination showed that the positive relationship between rural population aging and agricultural productivity gradually weakened as the degree of rural population aging deepened. The study provides empirical evidence for sustainable agricultural development in the context of population aging.

**Keywords:** aging; agricultural development; capital deepening; agricultural labor productivity

## 1. Introduction

The important development concept of sustainable development has evolved since its introduction, and today sustainable development implies a balance of three pillars, one of which is the economic sustainability necessary to maintain the natural, social, and human capital needed to sustain income and living standards [1,2]. As a large agricultural country, China's sustainable agricultural development is the basis for ensuring the development of the national economy and necessary for achieving long-term economic development. In recent years, the impact of the COVID-19 pandemic on food security has made agriculture the weakest link in the national economy, and the development of agriculture urgently needs new engines. The 14th Five-Year Plan focuses on the priority development of agriculture and proposes to promote the increase in farmers' income, the increase in industrial efficiency, and the increase in ecological value, promoting the transformation of agriculture from increasing production to high-quality development, further clarifying the goals and tasks of future agricultural development. It can be seen that high-quality agricultural development has become an important guiding direction for agricultural development. The essence of high-quality agricultural development is the improvement of productivity, especially the improvement of agricultural labor productivity [3].

Labor is the most direct and critical factor affecting the improvement of agricultural productivity. However, with the rapid advancement of China's new urbanization and the continuous decline in the proportion of agricultural labor, rural population aging has become an irreversible fact in the development process of the primary industry [4]. Data from the seventh national population census showed that in 2020, the population aged 60

and above in rural areas accounted for 23.8% of the total rural population, while in urban areas, it was 15.8%, a difference of 8 percentage points, which was significantly wider than the 4.3 percentage points in 2015. It is worth considering what kind of relationship exists between rural population aging and agricultural labor productivity, how it plays a role, and whether it can be verified using China's data.

Many previous studies attempted to conduct in-depth research on the relationship and causal pathways between aging and agricultural development, but the conclusions are still not clear. Some studies have found that population aging will constrain agricultural development. Feder [5] found that aging led to an increase in medical expenses, pension expenses, and other expenses in family expenditure, a decrease in capital investment in crop production, and constraints on agricultural development. Munnich et al. [6] found that elderly people engaging in agriculture not only affected the progress of agricultural production but also hindered the promotion and application of advanced agricultural technologies and tools due to the contrast between the difficulty in accepting modern agricultural technology progress and traditional ways of life, thereby restricting the improvement of production efficiency. Wang and Li [7] found that rural population aging means that the number of entities engaged in agricultural production activities is gradually decreasing, and the quality of entities is gradually aging, which will lower agricultural labor productivity and have a negative impact on agricultural production. Other studies have found that aging will not bring adverse effects to agricultural development. Ronald and Andrew [8] analyzed 115 countries from 2006 to 2010 and found that as the aging process deepens, the shortage of agricultural labor supply will force technological progress and improved productivity. Gao and Deng [9] found that rural population aging will accelerate the mechanization process, thereby promoting the improvement of wheat production efficiency and increasing wheat yield through empirical analysis. Liu [10] found that there was a significant positive relationship between rural population aging and total factor productivity in agriculture, and this effect was mainly caused by human capital accumulation. It can be seen that the research conclusions between aging and agricultural development are still unclear.

After reviewing the literature, it was found that more research has focused on the impact of aging on agricultural production and total factor productivity [11–13]. Agricultural production mainly refers to changes in the value added to the primary industry, which is a measure of total change. Agricultural total factor productivity is the residual after the direct contributions of capital and labor are removed, which measures the contribution of agricultural technological progress to agricultural production. Agricultural labor productivity is the output per unit of labor input on average, which is the core indicator for measuring agricultural development. Based on panel data from 30 provinces in China from 1999 to 2020, this article directly studied the relationship between rural aging and agricultural development from the perspective of labor productivity, using a fixed effects model to enrich the value research of population transition and agricultural development. Secondly, considering that capital deepening is closely related to agricultural labor productivity, this article explored the mechanism of the impact of rural aging on agricultural development using capital deepening as a mediating variable, providing empirical evidence and academic support for the government to deepen its promotion of high-quality agricultural development in the context of aging.

## 2. Research Hypotheses

Currently, there are still members of the labor force in rural China. Moderate aging and labor force transfer can effectively alleviate the problem of the "excessive density" of agricultural labor and bring positive effects to agricultural development [14]. Secondly, Sun [15] pointed out that elderly people can adjust the agricultural production structure and methods according to their own labor levels. Reasons such as low market competitiveness due to age, physical strength, and cultural level restrictions will prompt some young elderly people (aged 60–69) in rural areas to return to their hometowns and engage in

agricultural production activities again [16]. Young elderly people who have been engaged in agricultural production activities for many years have rich practical experience in agriculture and can carry out relevant agricultural production activities according to the growth cycle of crops, and their agricultural production experience and technical proficiency can produce a cumulative effect of technical experience, thereby improving labor productivity. As they age, the agricultural labor capacity of middle-aged and elderly people (aged 70–79) gradually declines, which also prompts them to replace manual labor with simple and lightweight machinery and equipment to carry out heavy and intensive agricultural labor, which improves labor productivity to a certain extent [17,18]. Furthermore, aging can also promote middle-aged and elderly people to transfer land to professional farmers for production management, which will inevitably improve agricultural labor productivity. Finally, in rural areas, the average life expectancy is 72.43 years old, and the proportion of elderly people aged 80 and above is only 2.4%, and the probability of complete disability at the age of 80 is only 15.5%. It can be seen that in the aging stage, the elderly population in rural areas still has a strong labor capacity. Based on the above analysis, this article proposes the following hypothesis:

**H1:** *There is a positive relationship between the aging of the rural population and agricultural labor productivity.*

Capital deepening mainly refers to the process of the amount of capital per capita increase over time, and the growth rate of capital factors far exceeds that of labor factors. Capital deepening can effectively promote agricultural productivity [19] and is a natural response of agriculture to urbanization and population aging [20]. Rural population aging means that the elderly population in rural areas is gradually increasing, and the total rural labor force is decreasing, which will force a change in the element structure of agriculture. On the one hand, aging will cause labor factor prices to rise, and in order to reduce agricultural production costs, it will stimulate agricultural production subjects to release their own labor force and increase capital input to replace labor input and ultimately improve agricultural labor productivity [21]. On the other hand, rural population aging can accelerate the continuous input of capital, services, and other factors in the agricultural sector. The inflow of a large amount of capital can not only force the upgrading of agricultural technology, but also bring about scale agriculture and efficient agriculture, promoting the transformation of agricultural production methods from extensive to intensive development, and ultimately improving agricultural labor productivity [22]. Based on this, this article proposes the following hypothesis:

**H2:** *Rural population aging will accelerate the accumulation of capital elements in the agricultural sector and ultimately improve agricultural labor productivity.*

## 3. Research Design

*3.1. Model Construction*

In order to reveal the relationship between rural population aging and agricultural labor productivity, this paper used multiple measures to bridge the gap between different models while ensuring the robustness of the findings. A simple OLS regression was first applied and the model was constructed as follows:

$$labor_{it} = \alpha_0 + \alpha_1 age_{it} + \varepsilon_{it} \tag{1}$$

In model (1), labor is the agricultural labor productivity, age is the rural population aging, $\alpha_0$ is the constant term, $\alpha_1$ are the regression coefficients, and $\varepsilon$ is the random disturbance term.

The simple OLS regression ignores the influence of individual effects and time effects on the regression results. This paper further constructed a two-way fixed effects model to control for province fixed effects and time fixed effects and eliminate the influence of differences in economic development and natural endowments between samples due to

locational differences and time dynamics. At the same time, control variables were added to reduce the error caused by the impact of other factors on labor productivity. The model was set as follows:

$$labor_{it} = \beta_0 + \beta_1 age_{it} + \beta_2 x_{it} + \mu_j + \gamma_t + \varepsilon_{it} \qquad (2)$$

In model (2), $x$ is the control variable (including agricultural production structure, rural population quality, natural disaster degree, agricultural financial investment, urban-rural income gap, urbanization rate). $\beta_1$ and $\beta_2$ are parameters to be estimated, $\mu$ is the individual fixed effect, $\gamma$ is the time fixed effect, and the remaining variables are consistent with the above.

*3.2. Variable Description*

The dependent variable is agricultural labor productivity (labor). Improving agricultural labor productivity is the fundamental measure to achieve high-quality development of agriculture. This article refers to the research of Fan [23] and used the ratio of added value of the primary industry to the number of employees in the primary industry to represent labor productivity. To ensure the comparability of data, the actual agricultural value added of the agricultural sector was obtained by subtracting the primary industry value added of each province in 1998.

The independent variable is rural population aging (age). In empirical studies, indicators such as the proportion of the elderly population to the total population, and the elderly dependency ratio are generally used to measure the degree of aging. Based on the international definition of aging, this article chose the proportion of rural population aged 65 and above in the total rural population of each province to measure the rural aging status.

The intermediary variable is capital deepening (capital). For the agricultural production sector, capital deepening means that agricultural workers promote more capital in the work process, which is an important factor in improving agricultural labor productivity [24]. Many existing studies measured the capital/labor ratio. Therefore, this article chose the ratio of capital investment in agriculture, forestry, animal husbandry, and fishery to the number of employed persons in agriculture, forestry, animal husbandry, and fishery to measure the degree of capital deepening in the agricultural sector. Since fixed asset formation data are not publicly available in some provinces' statistical yearbooks, the measurement of agricultural capital investment adopted the perpetual inventory method to process the total fixed asset investment of agriculture, forestry, animal husbandry, and fishery [25].

In order to control for the impact of other factors on agricultural labor productivity, this article selected the following control variables [26–29]:

1. Agricultural production structure (struct). This was represented by the proportion of grain sowing area to total crop sowing area in each region.
2. Rural population quality (quality). Generally speaking, the quality of rural labor force can effectively promote the high-quality development of agriculture. This article used the proportion of the population with a high school education or above in rural areas to the total population to represent the rural population quality.
3. Degree of natural disasters (disaster). Natural factors are an important factor affecting agricultural production. This article used the proportion of disaster-stricken area to crop sowing area to represent the degree of natural disasters.
4. Agricultural financial input (input). In order to support agricultural development, the government invests more funds in agricultural production. Therefore, this article used the expenditure on agriculture, forestry, and water affairs to represent the government's financial input into agriculture.
5. Urban–rural income gap (gap). This variable can significantly affect the production decision-making behavior of agricultural practitioners. A smaller income gap means

that farmers have more ability and capital elements to invest in agricultural production, thereby improving productivity. This article used the ratio of per capita disposable income of urban and rural residents to represent the urban–rural income gap.

6. Urbanization rate (urban). Urbanization provides strong support for agricultural development in terms of funds, technology, talents, and markets, which is conducive to improving agricultural labor productivity. This article used the proportion of urban population in each province to represent the urbanization rate.

The descriptive statistics of the variables are shown in Table 1. The table shows the variable definitions, observed values, mean, and standard deviation of each variable.

**Table 1.** Descriptive Statistics of the Variables.

| Variable | Variable Definition | Observation | Mean | Std. Dev. |
|---|---|---|---|---|
| Agricultural labor productivity | ln (added value of primary industry/number of employees in primary industry) | 630 | 0.562 | 9.073 |
| Rural population aging | the percentage of rural population aged 65 and above (%) | 630 | 9.897 | 3.164 |
| Capital deepening | ln (capital input in agriculture, forestry, animal husbandry, and fishery/number of employees in agriculture, forestry, animal husbandry, and fishery) | 630 | 1.11 | 7.994 |
| Agricultural production structure | the percentage of grain planting area in the total crop planting area (%) | 630 | 69.87 | 68.019 |
| Rural population quality | the percentage of population with high school education or above in rural areas (%) | 630 | 4.543 | 8.705 |
| Natural disaster degree | the percentage of disaster-affected area in the crop planting area (%) | 630 | 17.9 | 20.93 |
| Agricultural fiscal input | ln (expenditure on agricultural, forestry, and water affairs) | 630 | 1.434 | 5.068 |
| Urban–rural income gap | the ratio of per capita disposable income of urban residents to that of rural residents (in CNY/person) | 630 | 0.564 | 2.734 |
| Urbanization rate | the percentage of urban population in the total population (%) | 630 | 15.655 | 49.245 |

*3.3. Data Sources*

The data used in this article are from the "China Population and Employment Statistics Yearbook", the "China Rural Statistics Yearbook", and the "China Statistical Yearbook". Interpolation was used to supplement data with a few missing values, and samples from Tibet with severe missing data were excluded. Considering the time point of entering aging and the limitations of data acquisition, panel data from 30 provinces in China from 1999 to 2020 were selected to explore the relationship and mechanism between rural population aging and agricultural development.

## 4. Empirical Results and Analysis

*4.1. Baseline Regression Results*

According to the research design in the previous section, this article used fixed effects and OLS methods to simultaneously conduct regression. The specific results are shown in Table 2. Columns (1) and (2) only examined the relationship between rural population aging and agricultural labor productivity. Columns (3) and (4) further added some control

variables. The regression coefficients of aging were all significantly positive, indicating the reliability of the regression results. The regression results in column (4) show that the coefficient value of rural population aging was 0.046, and it was significantly positive at the 1% statistical level. This indicates that there was a positive relationship between rural population aging and agricultural labor productivity, but whether there was a mediating effect still needs further study. After controlling for variables that affect agricultural labor productivity, the coefficient of rural population aging was still significantly positive. The coefficient value decreased from 0.127 before controlling to 0.046, which may be due to the control variables absorbing the relevant effects. Overall, rural population aging and agricultural labor productivity had a significant positive relationship, which verifies Hypothesis 1.

**Table 2.** Regression Results of Population Aging and Agricultural Labor Productivity.

| Variable | (1) OLS | (2) FE | (3) OLS | (4) FE |
|---|---|---|---|---|
| age | 0.094 *** | 0.127 *** | 0.018 *** | 0.046 *** |
| | (0.000) | (0.000) | (0.001) | (0.000) |
| struct | | | 0.000 *** | 0.000 *** |
| | | | (0.757) | (0.388) |
| quality | | | 0.018 *** | 0.0296 *** |
| | | | (0.000) | (0.000) |
| disaster | | | −0.003 *** | −0.0017 *** |
| | | | (0.003) | (0.000) |
| input | | | 0.001 *** | 0.001 |
| | | | (0.000) | (0.913) |
| gap | | | −0.380 *** | −0.062 *** |
| | | | (0.000) | (0.003) |
| urban | | | 0.011 *** | 0.015 *** |
| | | | (0.000) | (0.000) |
| Constant | 8.099 *** | 7.778 *** | 9.252 *** | 7.768 *** |
| | (0.000) | (0.000) | (0.000) | (0.000) |
| Observations | 630 | 630 | 630 | 630 |
| R-squared | 0.281 | 0.683 | 0.673 | 0.852 |
| F value | 276.825 | 104.854 | 260.481 | 100.133 |

Note: *** indicates significant at the 1% level of statistical significance. Values in parentheses are *p*-values.

In terms of controlling for variable regression coefficients, the regression coefficient for agricultural production structure was significantly positive, indicating that the continuous optimization of agricultural production structure can promote the improvement of agricultural labor productivity. The regression coefficient for rural population quality was significantly positive, indicating that the improvement of rural population quality can solve actual problems in agricultural production with advanced technology and thus improve agricultural productivity. The regression coefficient for agricultural fiscal input was positive but not significant, indicating that although increasing funding to support agricultural development can improve agricultural labor productivity, the impact was relatively limited. The regression coefficient for urbanization rate was significant and positive, indicating that increasing the urbanization rate can effectively promote the improvement of agricultural labor productivity. However, frequent natural disasters have severely limited the improvement of agricultural labor productivity, and the regression coefficient for the natural disaster index was significantly negative. The regression coefficient for the income gap between urban and rural residents also showed a significant negative value, indicating that the widening income gap between urban and rural residents will hinder the improvement of agricultural labor productivity.

*4.2. Robustness Check*

Based on the research results above, there was a significant positive correlation between rural population aging and agricultural labor productivity. In order to enhance the reliability of the research conclusions, this article conducted a robustness check, and the specific results are shown in Table 3.

**Table 3.** Robustness Results.

| Variable | (1) (Substitution of the Explanatory Variables) | (2) (Substitution of the Explained Variable) | (3) (Endogeneity Test) |
|---|---|---|---|
| age | 0.003 ** | 0.033 *** | 0.267 ** |
| | (0.013) | (0.000) | (0.011) |
| Constant | 8.972 *** | 8.284 *** | 7.876 ** |
| | (0.000) | (0.000) | (0.028) |
| _cons | YES | YES | YES |
| Individual FE | YES | YES | YES |
| Year FE | YES | YES | YES |
| Observations | 630 | 630 | 630 |
| R-squared | 0.712 | 0.896 | 0.531 |
| F value | 107.982 | 131.442 | 49.684 |

Note: **, *** indicate significant at the 5% and 1% levels of statistical significance, respectively. Values in parentheses are *p*-values.

4.2.1. Substitution of Explanatory Variables

Following the method of Fan et al. [30], the measurement index of the core explanatory variable was replaced with the rural elderly dependency ratio, which is the ratio of the rural population aged 65 and above to the rural population aged 15 to 64. According to the results in column (1), it can be found that after replacing the core explanatory variable, the direction and significance level of the regression coefficient of the explanatory variable were basically consistent with the benchmark regression results, indicating that the relationship between rural population aging and agricultural labor productivity was very stable, and the conclusions obtained earlier are reliable.

4.2.2. Substitution of the Explained Variable

Following the research of Wang et al. [31], we used the ratio of the total output value of the primary industry to the number of employees in the primary industry in 1998 as a measure of agricultural labor productivity. The results in column (2) show that after replacing the explained variable, the coefficient value of rural population aging changed from 0.046 in the benchmark regression to 0.033, and it was significant at the 1% level, indicating that there was still a significant positive correlation between rural population aging and agricultural labor productivity, which enhances the reliability of the research conclusions.

4.2.3. Considering Endogeneity Issues

Endogeneity is an unavoidable issue in the empirical testing process of a model, and its existence will lead to inconsistencies in parameter estimations. Therefore, in order to alleviate the endogeneity problem in the empirical process, this article used a panel dynamic GMM model with lagged agricultural labor productivity to conduct regression. The results in column (3) show that there was a significant positive correlation between rural population aging and agricultural labor productivity, and it passed the 5% significance test, which verifies the reliability of the research conclusions.

## 5. Mechanism Test and Heterogeneity Test

### 5.1. Testing the Mediating Effect Mechanism

The previous study showed that rural population aging can promote agricultural labor productivity, so what is the mechanism of its empowering agricultural labor productivity? To further reveal the deep mechanism of rural population aging to promote labor productivity, this paper referred to the study of Wen et al. [32] to establish an intermediary effect model for testing:

$$labor_{it} = a_1 + b_1 age_{it} + c_1 x_{it} + \mu_j + \gamma_t + \varepsilon_{it} \tag{3}$$

$$capital_{it} = a_2 + b_2 age_{it} + c_2 x_{it} + \mu_j + \gamma_t + \varepsilon_{it} \tag{4}$$

$$labor_{it} = a_3 + b_3 age_{it} + b_4 capital_{it} + c_3 x_{it} + \mu_j + \gamma_t + \varepsilon_{it} \tag{5}$$

Capital is the mediating mechanism variable in model (4) and model (5), and the coefficient $b_2$ of age in model (4) indicates the effect of the aging of the rural population on the mediating mechanism variable. Among them, the coefficient of age $b_2$ and the coefficient of the mediating variable capital $b_4$ are the core coefficients of interest in this paper, and if they are both significant, they indicate that the aging of the agricultural population promotes labor productivity improvement with the mediating mechanism variable as the transmission mechanism. The rest of the variables are consistent with the previous definitions. The previous theoretical analysis showed that rural population aging can accelerate capital accumulation, effectively improving the level of agricultural labor productivity by promoting modernization of production methods. Referring to the existing research ideas, this article used the mediation effect model designed in the previous section to verify this, and the specific results are shown in Table 4. The regression results in column (2) show that the coefficient value of rural population aging was significantly positive at the 1% level of statistical significance, indicating that aging significantly promoted capital deepening. At the same time, as shown in column (3) of Table 3, the coefficient values for rural population aging and capital deepening were both significantly positive at the 1% level of statistical significance, which means that capital deepening did play a partial mediating role between rural population aging and agricultural labor productivity, consistent with the theoretical expectations in the previous section.

**Table 4.** Regression Results of the Mediating Effect Mechanism.

| Variable | (1) (Labor) | (2) (Capital) | (3) (Labor) |
|---|---|---|---|
| age | 0.046 *** (0.000) | 0.063 *** (0.000) | 0.020 *** (0.000) |
| capital | | | 0.427 *** (0.000) |
| Constant | 7.768 *** (0.000) | 4.811 *** (0.000) | 5.715 *** (0.000) |
| _cons | YES | YES | YES |
| Individual FE | YES | YES | YES |
| Year FE | YES | YES | YES |
| Observations | 630 | 630 | 630 |
| R-squared | 0.852 | 0.797 | 0.741 |
| F value | 100.135 | 262.792 | 206.601 |

Note: *** indicates significant at the 1% level of statistical significance. Values in parentheses are *p*-values.

### 5.2. Heterogeneity Test

#### 5.2.1. Regression Test by Time Period

Using the entire sample for regression analysis may obscure the relationship between rural population aging and agricultural labor productivity in different time periods. There-

fore, taking 2010 as the node, this study analyzed whether there was a significant change in the relationship between rural population aging and agricultural labor productivity before and after 2010. The specific results are shown in Table 5. The results show that the coefficient value of rural population aging from 1999 to 2009 was 0.04, and the significance level did not exceed 1%. From 2010 to 2020, the coefficient value of rural population aging was positive and relatively significant (0.038). Comparing the two, although the positive relationship between rural population aging and agricultural labor productivity still existed, the strength of the relationship was gradually weakening, and the dependence of agricultural production on labor was decreasing.

**Table 5.** Heterogeneity Regression Results.

| Variable | (1)<br>(1999–2009) | (2)<br>(2010–2020) | (3)<br>(Low Aging<br>Group) | (4)<br>(Medium<br>Aging Group) | (5)<br>(High Aging<br>Group) |
|---|---|---|---|---|---|
| age | 0.040 *** | 0.038 *** | 0.139 *** | 0.032 *** | 0.009 |
|  | (0.000) | (0.023) | (0.000) | (0.000) | (0.123) |
| Constant | 7.587 *** | 7.638 *** | 7.724 *** | 7.760 *** | 5.884 *** |
|  | (0.000) | (0.000) | (0.000) | (0.000) | (0.000) |
| _cons | YES | YES | YES | YES | YES |
| Individual FE | YES | YES | YES | YES | YES |
| Year FE | YES | YES | YES | YES | YES |
| Observations | 330 | 300 | 98 | 469 | 63 |
| R-squared | 0.821 | 0.799 | 0.895 | 0.878 | 0.931 |
| F value | 124.850 | 114.774 | 92.934 | 205.919 | 83.817 |

Note: *** indicates significant at the 1% level of statistical significance. Values in parentheses are *p*-values.

### 5.2.2. Regression Test by Aging Degree

In order to explore whether there was a significant change in the relationship between rural population aging and agricultural labor productivity under different degrees of rural population aging, this study divided the entire sample into three groups, low, medium, and high, based on whether the aging degree exceeded 7% and 14%, and conducted a heterogeneity test. Samples with a less than 7% aging degree were classified as the low aging group, 7% to 14% were classified as the medium aging group, and samples with a more than 14% aging degree were classified as the high aging group. The specific results are shown in Table 5. The results show that in the low and medium aging groups, the rural population aging variable was significantly positive at the 1% statistical level. However, in the high aging group, the coefficient value of the variable was 0.009, but it was not significant. It can be seen that the relationship between rural population aging and agricultural labor productivity was closely related to the aging degree. When the aging degree of rural population aging was low, it showed a significant positive relationship with agricultural labor productivity. As the aging degree of the rural population aging increased, the positive relationship with agricultural labor productivity gradually weakened.

### 6. Discussion

As a large agricultural development country with a large population, the impact of population aging on China's agricultural development has become an unavoidable research focus. The issue of sustainable agricultural development caused by changes in the age structure of the agricultural population has become a hot topic of current discussion, and it is of great practical importance to identify the mechanism of the impact of the aging of the rural population on agricultural labor productivity for the long-term sustainable development of China's agriculture. In this paper, we used data from 30 Chinese provinces to verify whether the aging of the agricultural population has an impact on agricultural labor productivity. Based on the test results, we found that there was a significant positive relationship between the aging of the rural population and agricultural labor productivity, but this relationship was gradually weakening with the passage of time and the deepening

of aging. The findings of this paper enrich the research on the impact of population aging on productivity in agriculture in the Chinese context, and have important theoretical and practical implications for improving agricultural economic dynamics and promoting sustainable agricultural development.

On the one hand, a review of the existing literature showed that the impact of rural population aging on agricultural development has received widespread attention from the academic community, but the research findings are mixed. Taking agricultural labor productivity, a core indicator of agricultural development, as the core explanatory variable, this paper explored the impact of an aging agricultural population on agricultural labor productivity and the mechanistic role of capital deepening in it, enriching the research on demographic transitions and agricultural development and providing direct evidence on the impact of demographic change on agricultural productivity. On the other hand, it provides an important reference for the scientific decision making of the Chinese government, which can strengthen the ability of government departments to cope with the impact of demographic changes on agriculture and promote high-quality agricultural development in the future. This study suggests that the Chinese government should give full play to the role of local vocational education to cultivate agricultural talents based on the national and agricultural conditions of China's aging population. It should guide the transfer of capital factors to agriculture through fiscal and tax policies, improve the efficiency of agricultural resource allocation, and promote the long-term sustainable development of agriculture by positive means.

## 7. Conclusions and Policy Implications

In recent years, the aging of rural populations has accelerated, and the agricultural economy is undergoing a transformation towards high-quality development. The importance and urgency of improving agricultural labor productivity are becoming increasingly apparent. Using panel data from 30 provinces in China from 1999 to 2020, this paper examined the relationship between the aging of the rural population and agricultural development from the perspective of labor productivity, and explored the underlying mechanisms. The study found that there was a significant positive relationship between the aging of the rural population and agricultural labor productivity, but this relationship gradually weakened over time and as the degree of aging increased. Through further mechanistic analysis, it was found that the aging of the rural population can drive the continuous accumulation of factors such as capital and technology, thereby improving agricultural production methods and ultimately realizing the improvement of agricultural labor productivity.

Based on the above research findings, and in the context of the aging of the rural population becoming an established fact and continuing to deepen, this paper proposes the following suggestions to improve agricultural labor productivity:

First, open up channels for capital, technology, and other factors to flow from cities to rural areas, and guide these factors to transfer to agriculture. The government can use policy measures such as fiscal investment, tax breaks, and green finance to encourage social capital to invest in agricultural production, improve agricultural infrastructure, enhance the technological elements of agricultural industrial development, replace labor with capital, promote the professionalization, scale, and modernization of agriculture, and ultimately realize high-quality agricultural development.

Second, we should develop vocational education and cultivate new types of professional farmers. Due to regional differences in natural environmental endowments, agricultural production using empirical knowledge and traditional production factors is an important way of agricultural development in China, while an aging agricultural population has led to the transfer of some land into the hands of full-time farmers with agricultural experience. China's unique agricultural production resource endowment and the status quo of smallholder-oriented agricultural operations dictate that China cannot abandon traditional agricultural production methods and adopt fully modernized agri-

cultural production methods. Therefore, the accumulation of knowledge of traditional agricultural production experience is still extremely relevant to the development of agricultural modernization at this stage. The government should improve the quality of new professional farmers at the grassroots level through incentive and guidance policies, and give full play to the role of local vocational education to cultivate agricultural talents.

Third, improve agricultural production technology, especially the promotion of digital technology in agricultural production. Currently, technological progress in agriculture is a key factor affecting the improvement of agricultural labor productivity. Therefore, the government should vigorously promote the transformation of traditional agriculture into modern agriculture, and accelerate the realization of high-quality agricultural development.

**Author Contributions:** Conceptualization, D.S.; methodology and software, H.L.; writing—original draft preparation, H.L. and W.S.; writing—review and editing, D.S. and W.S. All authors have read and agreed to the published version of the manuscript.

**Funding:** The project was supported by the Shanxi Provincial Government Major Decision-making Consulting Project (No. 2020ZB08007) and North University of China Postgraduate Science and Technology Project (No. 20221850).

**Informed Consent Statement:** Not applicable.

**Data Availability Statement:** Partial data openly available in a public repository. The data that support the findings of this study are openly available in https://www.stats.gov.cn/. Data available on request from the authors. Partial data that support the findings of this study are available from the author upon reasonable request.

**Acknowledgments:** The authors would like to thank the editor and the anonymous referees for their helpful comments and suggestions.

**Conflicts of Interest:** The authors declare no conflict of interest.

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
