# Peer review of "Rural Population Aging, Capital Deepening, and Agricultural Labor Productivity"

_sustainability, doi:10.3390/su15108331_

Round 1

Reviewer 1 Report

The study is scientifically sound although the policy recommendations seem to be in some case too utopic and impossible to implement. There are some terms that need to be explained to increase clarity and quality of the findings.

Reviewer 2 Report

This is a well written paper, with some robust argumentation and interesting findings.

Nevertheless, I have the following major concerns:

1.     Given the journal’s scope, the authors should place the whole discussion under the notion of sustainable development. Thus, a short discussion of the term should be provided in the introduction. In this vein, the following two papers should be included. (a) Manioudis, M. & Meramveliotakis, G. (2022) “Broad strokes towards a grand theory in the analysis of sustainable development: a return to the classical political economy”, New Political Economy, 27(5), pp. 866-878, and (b) Tomislav, K. (2018) “The concept of sustainable development: From its beginning to the contemporary issues”, Zagreb International Review of Economics & Business, 21(1), 67-94.

2.     A discussion section should be included before Conclusions.

3.     Authors reside exclusively to aging population the cause for new technology insertion in the agricultural section. I am not sure if this claim can hold, given that the use of new technology in any sector seems to be a deterministic law in the contemporary capitalism. In plain terms, what I am arguing is that assuming that there is no aging population in rural areas in China, the insertion of new technological methods of production is still driven by the structural transformation of agriculture in to the lens of capitalist production.

Reviewer 3 Report

 Abstract is not self-explanatory – methodology is missing;

Introduction: References cited in lines 45 to 67 used different methodologies and variables, thus is difficult to compare them;

Research hypothesis – it is not clear how the two research hypothesis were established;

Model construction – limits of the model are not described; it is not clear how the hypothesis of the paper is introduced in the model.

Variable described in table 1 are not compared with other similar studies;

Discussions are missing.

Round 2

Reviewer 2 Report

Glad to see the authors accommodated all of my comments.

I suggest publication.

Reviewer 3 Report

The authors change the paper accordingly to the suggestions.